# Skinned Motion Retargeting with Dense Geometric Interaction Perception

**Zijie Ye**[1,2]*, **Jia-Wei Liu**[3], **Jia Jia**[1,2]†, **Shikun Sun**[1,2], **Mike Zheng Shou**[3]

[1] Department of Computer Science and Technology, BNRist, Tsinghua University
[2] Key Laboratory of Pervasive Computing, Ministry of Education
[3] Show Lab, National University of Singapore

## Abstract

Capturing and maintaining geometric interactions among different body parts is crucial for successful motion retargeting in skinned characters. Existing approaches often overlook body geometries or add a geometry correction stage after skeletal motion retargeting. This results in conflicts between skeleton interaction and geometry correction, leading to issues such as jittery, interpenetration, and contact mismatches. To address these challenges, we introduce a new retargeting framework, *MeshRet*, which directly models the dense geometric interactions in motion retargeting. Initially, we establish dense mesh correspondences between characters using semantically consistent sensors (SCS), effective across diverse mesh topologies. Subsequently, we develop a novel spatio-temporal representation called the dense mesh interaction (DMI) field. This field, a collection of interacting SCS feature vectors, skillfully captures both contact and non-contact interactions between body geometries. By aligning the DMI field during retargeting, *MeshRet* not only preserves motion semantics but also prevents self-interpenetration and ensures contact preservation. Extensive experiments on the public Mixamo dataset and our newly-collected *ScanRet* dataset demonstrate that *MeshRet* achieves state-of-the-art performance. Code available at https://github.com/abcyzj/MeshRet.

## 1 Introduction

Skinned character animation is prevalent in virtual reality [16], game development [21], and various other fields. However, animating these characters often presents significant challenges due to differences in body proportions between the motion source and the target character, leading to issues such as loss of motion semantics, mesh interpenetration, and contact mismatches. Consequently, motion retargeting is essential to adjust for these discrepancies in body proportions. This process is crucial for maintaining the integrity of the source motion's characteristics in the animation of the target character.

Motion retargeting presents challenges due to the complex interactions among character limbs and the wide range of body geometries. Accurately preserving these interactions is crucial, as incorrect interactions can result in mesh interpenetration and contact mismatches. Prior research has typically addressed these interactions from two perspectives: skeleton interactions and geometry corrections. Early methods [1, 29, 15] employ cycle-consistency to implicitly align skeleton interaction semantics, yet they do not address the complexities of geometric interactions between different body parts. Villegas et al. [28] introduced mesh self-contact modeling; however, their approach does not extend

---

*Work is partially done during visiting NUS.
†Corresponding Author.

38th Conference on Neural Information Processing Systems (NeurIPS 2024).

to non-contact interactions. More recently, Zhang et al. [32] implemented a two-stage pipeline that first aligns skeleton interaction semantics and then corrects geometric artifacts. Nonetheless, the inherent conflict between preserving skeleton interaction semantics and correcting geometry leads to jittery movements, severe interpenetration and imprecise contacts. Zhang et al. [30] subsequently proposed adding a stage that aligns visual semantics with a visual language model, but this requires detailed pair-by-pair finetuning due to the loss of spatial information when projecting 3D motion into 2D images.

To resolve the conflict between skeleton interaction and geometry correction, we propose a new approach: focusing solely on dense geometric interaction for motion retargeting. Character animation videos, rendered from the skinned mesh, rely on geometric interactions to shape user perception. Skeleton interaction, in contrast, merely represents a simplified, sparse form of geometric interaction. Therefore, maintaining correct interactions between different body part geometries not only preserves motion semantics but also prevents mesh interpenetration and ensures contact preservation, as illustrated in Figure 1.

Given the significance of geometric interactions, we propose a new framework, named *MeshRet*, for skinned motion retargeting. In contrast to earlier methods that adjust skeletal motion retargeting outcomes, our approach models the intricate interactions among character meshes without depending on predefined vertex correspondences.

The design of *MeshRet* necessitates several technical innovations. Initially, there is a requirement for dense mesh correspondence across different characters. Drawing inspiration from the medial axis inverse transform (MAIT) [22], we have devised a technique, termed semantically consistent sensors (SCS), to automatically derive dense mesh correspondence from sparse skeleton correspondence. This technique enables us to sample a point cloud of sensors on the mesh to represent each character. Following this, to illustrate dense mesh interaction between body parts, we employ interacting mesh sensor pairs, maintaining generality. These pair-wise interactions are encoded within a novel spatial-temporal representation termed the Dense Mesh Interaction (DMI) field. The DMI field adeptly encapsulates both contact and non-contact interaction semantics. Finally, we proceed to learn a motion manifold that aligns with the target character geometry and the source motion DMI field.

To align our evaluation process more closely with real animation production, we gathered an in-the-wild motion dataset, termed *ScanRet*, characterized by abundant contact semantics and minimal mesh interpenetration. *ScanRet* consists of 100 human actors ranging from bulky to skinny, each performing 83 motion clips scrutinized by human animators. The *MeshRet* model is trained on both the *ScanRet* dataset and the widely used Mixamo [2] dataset. We assessed our method across a large variety of motions and a diverse array of target characters. Both qualitative and quantitative analyses show that our *MeshRet* model significantly outperforms existing methods.

To summarize, we present the following contributions:

- We introduce *MeshRet*, a pioneering solution that facilitates geometric interaction-aware motion retargeting across varied mesh topologies in a single pass.

- We present the SCS and the novel DMI field to guide the training of *MeshRet*, effectively encapsulating both contact and non-contact interaction semantics.

- We develop *ScanRet*, a novel dataset specifically tailored for assessing motion retargeting technologies, which includes detailed contact semantics and ensures smooth mesh interaction.

- Our experiments demonstrate that *MeshRet* delivers exceptional performance, marked by accurate contact preservation and high-quality motion.

## 2   Related Work

**Skeletal motion retargeting**   Motion retargeting seeks to preserve the characteristics of source motions when transferring them to a different target character. Skeletal motion retargeting primarily addresses the challenge of differing bone ratios. Gleicher [8] initially formulated motion retargeting as a spatio-temporal optimization problem, using source motion features as kinematic constraints. Subsequent researches [5, 7, 14] have focused on optimization-based approaches with various constraints. However, these methods, while requiring extensive optimization, often yield suboptimal

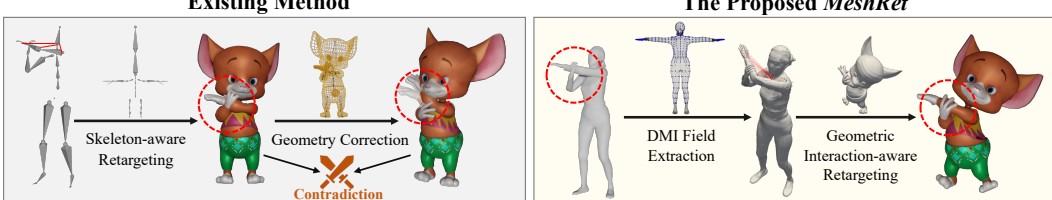

Figure 1: Comparison with the existing method. Contrary to the earlier retargeting-correction approach [32], which suffer from internal contradictions leading to interpenetration, jitter, and contact mismatches, our pipeline leverages the DMI field to accurately model complex geometric interactions.

results. Consequently, recent studies have explored learning-based motion retargeting algorithms. Jang et al. [11] trained a motion retargeting network using a U-Net [26] architecture on paired motion data. Villegas et al. [29] introduced a recurrent neural network combined with cycle-consistency [35] for unsupervised motion retargeting. Lim, Chang, and Choi [15] propose to learn frame-by-frame poses and overall movements separately. Aberman et al. [1] develop differentiable operators for cross-structural motion retargeting among homeomorphic skeletons. However, these methods generally neglect the geometry of characters, leading to frequent contact mismatches and severe mesh interpenetrations.

**Geometry-aware motion retargeting**   Previous studies have generally processed character geometries through two approaches: contact preservation and interpenetration avoidance. Lyard and Magnenat-Thalmann [19] developed a heuristic optimization algorithm to maintain character self-contact, while Ho, Komura, and Tai [9] proposed to maintain character interactions by minimizing the deformation of interaction meshes. Ho and Shum [10] introduced a spatio-temporal optimization framework to prevent self-collisions in robot motion retargeting. Jin, Kim, and Lee [12] employed a proxy volumetric mesh to preserve spatial relationships during retargeting. Subsequently, Basset et al. [4] combined both attraction and repulsion terms in an optimization-based method to avoid interpenetration and preserve contact. However, these methods necessitate per-vertex correspondence and involve costly optimization processes. More recently, Villegas et al. [28] attempted to retarget skinned motion through optimization in a latent space of a pretrained network, although their method does not accommodate non-contact interactions. Zhang et al. [32] implemented a two-stage pipeline that initially aligns skeleton interaction semantics and subsequently corrects geometric artifacts. Nevertheless, the inherent conflict between maintaining skeleton interaction semantics and correcting geometry often results in jittery movements and imprecise contacts. In a later study, Zhang et al. [30] added a stage that aligns visual semantics using a visual language model, but this approach requires extensive pair-by-pair fine-tuning due to the loss of spatial information when projecting 3D motion into 2D images.

Existing geometry-aware motion retargeting methods either require expensive optimization or employ multi-stage strategies for skeleton and geometry semantics, resulting in a contradiction between stages that often leads to unsatisfactory results. In contrast, our method processes both contact and non-contact semantics using a dense mesh interaction field in a single stage.

## 3   Method

### 3.1   Overview

We introduce a novel geometric interaction-aware motion retargeting framework *MeshRet*, as illustrated in Figure 2. Unlike previous methods that either overlook character geometries [1, 29, 15] or apply geometry correction after skeleton retargeting [32, 30], our framework directly addresses dense geometric interactions with the Dense Mesh Interaction (DMI) field. This provides a detailed representation of the interactions within skinned character motions, preserving motion semantics by preventing mesh interpenetration and ensuring precise contact preservation.

**Motion & geometry representations**   Assume the motion sequence has $T$ frames and the character has $N$ skeletal joints. The motion sequence $\mathbf{m}$ is represented by the global root translation $\mathbf{X} \in \mathbb{R}^{T \times 3}$

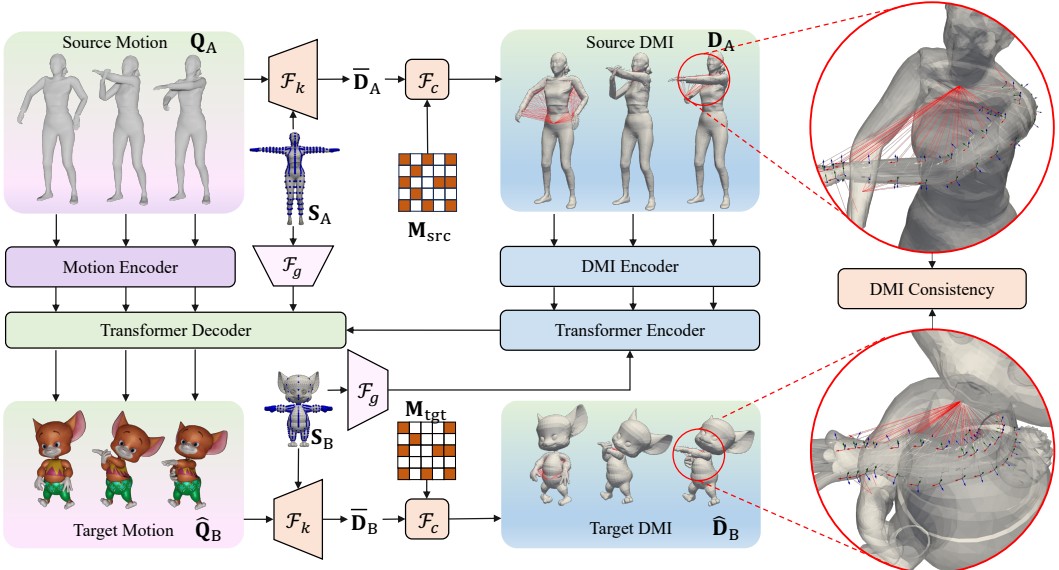

Figure 2: Overview of the proposed *MeshRet*. The pipeline begins with the extraction of the DMI field using sensor forward kinematics, denoted as $\mathcal{F}_k$, and pairwise interaction feature selection, represented by $\mathcal{F}_c$. This DMI field, in conjunction with geometric features derived from $\mathcal{F}_g$, is fed into an encoder-decoder network. The network predicts the target motion sequence, which is aligned with the target character's geometry and the original DMI field.

and the local joint rotation $\mathbf{Q} \in \mathbb{R}^{T \times N \times 6}$, where we adopt the 6D representation [34] for the joint rotations. The rest-pose geometry $\mathbf{G}$ of the character is represented by the rest-pose mesh $\mathbf{O}$ and the rest-pose joint locations $\mathbf{J} \in \mathbb{R}^{N \times 3}$.

**Task definition** Given the source motion sequence $\mathbf{m}_A$, and the geometries $\mathbf{G}_A$ and $\mathbf{G}_B$ of the source and target characters in their T-poses, our objective is to generate the motion $\mathbf{m}_B$ for the target character. This process aims to retain essential aspects of the source motion, including its semantics, contact preservation, and the avoidance of interpenetration.

Following the definition of the task, our *MeshRet* model initially derives Semantically Consistent Sensors (SCS) $\mathbf{S} \in \mathbb{R}^{S \times 4 \times 3}$, which provide dense geometric correspondences essential for the retargeting process, where $\mathbf{S} = \mathcal{F}_s(\mathbf{G})$. $\mathbf{S}$ captures the sensor location and the sensor tangent space matrix, facilitating an enhanced perception of the geometry surface. Subsequently, we conduct sensor forward kinematics (FK) and pairwise interaction extraction to generate the source DMI field $\mathbf{D}_A = \mathcal{F}_d(\mathbf{m}_A, \mathbf{S}_A)$, where $\mathbf{D}_A \in \mathbb{R}^{T \times K \times L \times P}$. Here, $K$ is the number of SCS in the DMI field, $L$ represents a hyper-parameter of feature selection, and $P$ indicates the feature dimension of the DMI. Lastly, a transformer-based network [27] ingests $\mathbf{m}_A$, $\mathbf{D}_A$, $\mathbf{S}_A$, and $\mathbf{S}_B$, and predicts a target motion sequence $\mathbf{m}_B$ that aligns with the target character's geometry and the source DMI field. The entire pipeline is denoted as follows:

$$\mathbf{m}_B = \mathcal{F}_r(\mathbf{m}_A, \mathbf{D}_A, \mathbf{S}_A, \mathbf{S}_B) \tag{1}$$

### 3.2 Semantically consistent sensors

To facilitate dense geometric interactions, our *MeshRet* framework necessitates establishing dense mesh correspondence between source and target characters. Previous studies have typically derived correspondence from vertex coordinates [33], virtual sensor [31] or through a bounding mesh [12]; however, these methods are confined to template meshes sharing identical topology, such as MANO [25] or SMPL [18]. Villegas et al. [28] suggested determining vertex correspondence using nearest neighbor searches on predefined feature vectors. Nevertheless, this approach often lacks precision and brevity, resulting in inaccurate contact representations and substantial optimization burdens.

In this study, we introduce Semantically Consistent Sensors (SCS) that are effective across various mesh topologies while ensuring precise semantic correspondence. Our approach draws inspiration

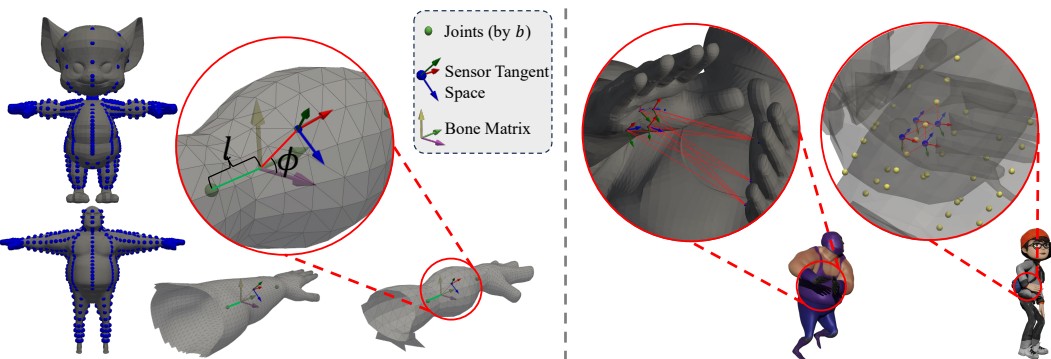

Figure 3: Left: Illustration of the method to derive a sensor feature $\mathbf{s}$ from the semantic coordinate $(b, l, \phi)$ across different characters. The red line represents the projected ray. The feature $\mathbf{s}$ encompasses the sensor's location and its tangent space matrix. Right: The DMI field effectively captures both contact and non-contact interactions. Red lines represent $\mathbf{d}^{t,i,j}$ in the DMI field. In the second example, the body sensors (yellow points) are located in the tangent plane of the hand sensors (blue points), signifying a contact interaction.

from the Medial Axis Inverse Transform (MAIT) [22]. We conceptualize the skeleton bones of each character as approximate medial axes of their limbs and torso. For each bone, a MAIT-like transform is applied to generate the corresponding SCS. This involves casting rays from the bone axis across a plane perpendicular to it. The origin parameter $l$ and direction parameter $\phi$ of the rays, combined with the bone index $b$, establish the semantic coordinates of the SCS. The semantic coordinates describe connection between the sensor and the skeleton bones. A sensor is deemed valid if its ray intersects the mesh linked to the bone; otherwise, it is considered invalid. Through this method, we establish a dense geometric correspondence based on sparse skeletal correspondence. The procedure for deriving SCS is illustrated in Figure 3. Given a unified set of SCS semantic coordinates $\{(b_1, l_1, \phi_1), (b_2, l_2, \phi_2), \cdots, (b_S, l_S, \phi_S)\}$, we can derive SCS feature $\mathbf{S} = \{\mathbf{s}_1, \mathbf{s}_2, \cdots, \mathbf{s}_S\}$ for each character. Further details can be found in Algorithm 1.

### 3.3 Dense mesh interaction field

To effectively represent the interactions between character limbs and the torso, we have developed the DMI field. Based on SCS detailed in Section 3.2, the DMI field comprehensively captures both contact and non-contact interactions across different body part geometries. Utilizing the DMI field allows for dense geometry interaction-aware motion retargeting, thereby eliminating the need for a geometry correction stage.

**Sensor forward kinematics** For a given motion sequence, denoted as $\mathbf{m}$, we initially conduct forward kinematics (FK) on $\mathbf{S}$ to derive sensor features $\mathbf{S}^{1:T} \in \mathbb{R}^{T \times S \times 4 \times 3}$. Each $\mathbf{S}^t$ encompasses the locations and tangent matrices for $S$ sensors at frame $t$. The FK transformation for an individual sensor is expressed as:

$$\mathbf{s}_i^t = \sum_{n=1}^{N} \omega(\mathbf{p}_i)_n G_n(\mathbf{Q}^t) \cdot \mathbf{s}_i, \tag{2}$$

where $G_n(\mathbf{Q}^t) \in SE(3)$ is the global transformation matrix for bone $n$, derived from its local rotation matrix, and $\omega(\mathbf{p}_i)_n$ represents the linear blend skinning (LBS) weight for sensor $\mathbf{s}_i$, determined through barycentric interpolation of its adjacent mesh vertices.

**Pairwise interaction feature** Next, we model the geometric interactions as pairwise interaction features between sensors. Ideally, for each frame, we obtain a comprehensive DMI field, $\overline{\mathbf{D}}^t$, representing pairwise vectors across $K^2$ sensor pairs:

$$\mathbf{d}^{t,i,j} = \mathbf{t}_i^{-1}(\mathbf{p}_j^t - \mathbf{p}_i^t), \tag{3}$$

$$\overline{\mathbf{D}}^t = \{(\mathbf{d}^{t,i,j}, b_i, b_j, l_i, l_j, \phi_i, \phi_j)\}_{i=1:S}^{j=1:S}, \tag{4}$$

where $\mathbf{t}_i \in \mathbb{R}^{3 \times 3}$ is the tangent matrix if sensor $i$, and $\mathbf{d}^{t,i,j}$ represents the relative position of target sensor $j$ in the tangent space of observation sensor $i$. $\overline{\mathbf{D}}^t$ is composed of two components: the relative position of the sensor pair and the semantic coordinates of both the observation and target sensors. The use of semantic rather than spatial coordinates is essential, as it obviates the need for actual sensor positions, thereby making DMI suitable for motion retargeting applications.

However, $\overline{\mathbf{D}}^t \in \mathbb{R}^{S \times S \times P}$ exhibits quadratic growth with respect to $S$ because it includes $S^2$ sensor pairs, rendering it impractical when managing thousands of sensors. To address this, we implement two sparsification strategies for $\overline{\mathbf{D}}^t$. Initially, we restrict interactions to critical body parts only, such as arm-torso, arm-head, arm-arm, and leg-leg, rather than between all sensor pairs, thereby restricting our focus to $K$ observation sensors. Subsequently, for each observation sensor, we select $L$ target sensors from each relevant body part, where $L$ is a predetermined hyper-parameter. Specifically, we empirically choose $L/2$ nearest and $L/2$ furthest target sensors. We find that proximate sensor pairs are crucial for minimizing interpenetration and maintaining contact, while distant pairs delineate the overall spatial relationships between body parts, as shown in Figure 3. These strategies lead to the formulation of the final DMI field $\mathbf{D} \in \mathbb{R}^{K \times L \times P}$, with selected sensor pairs indicated by the sparse DMI mask $\mathbf{M}_{\text{src}} \in \mathbb{R}^{S \times S}$ shown in Figure 2.

### 3.4 Geometry interaction-aware motion retargeting

To avoid the conflict between skeleton interaction and geometric correction, the proposed *MeshRet* employs the DMI field to model geometric interactions directly. As shown in Figure 2, *MeshRet* initially extracts the DMI field $\mathbf{D}_A$ from the source motion sequence $\mathbf{m}_A$, as described in Section 3.3. The field $\mathbf{D}_A$ encapsulates interactions among various body parts within the source motion, encompassing both contact and non-contact interactions, further depicted in Figure 3. The DMI field, composed of sensor pair feature vectors, possesses the unordered characteristics of a point cloud. Consequently, we implement a PointNet-like architecture [24] for our DMI encoder, which is divided into two components: the per-sensor encoder and the per-frame encoder. Given $\mathbf{D}_A \in \mathbb{R}^{T \times K \times L \times P}$, the per-sensor encoder initially processes it as $T * K$ separate point clouds, producing representations $\mathbf{H}_A^s \in \mathbb{R}^{T \times K \times D_{\text{model}}}$ for each observation sensor, where $D_{\text{model}}$ denotes the feature dimension. Subsequently, the per-frame encoder generates per-frame representations $\mathbf{H}_A^f \in \mathbb{R}^{T \times D_{\text{model}}}$ by encoding these $T$ point clouds.

Since DMI field $\mathbf{D}_A$ lacks geometric information about characters, we introduced a geometry encoder $\mathcal{F}_g$ to extract geometric features from their SCS. For each sensor, we form a feature vector by concatenating its rest-pose feature $\mathbf{s}_i$ with its semantic coordinates $(b_i, l_i, \phi_i)$. The resultant geometric features are represented as $\mathbf{C}_A \in \mathbb{R}^{S_A \times C}$ for character A and $\mathbf{C}_B \in \mathbb{R}^{S_B \times C}$ for character B. The semantic coordinates of sensors act as intermediaries linking the DMI field to character geometry. The geometry encoder employs a PointNet-like architecture [24] to transform the geometric features $\mathbf{C}$ into a geometric latent code $\mathbf{H}^g \in \mathbb{R}^{D_{\text{model}}}$.

The transformer-based retargeting network processes input features including the source DMI feature $\mathbf{H}_A^f$, source joint rotation $\mathbf{Q}_A$, source geometry latent $\mathbf{H}_A^g$, and target geometry latent $\mathbf{H}_B^g$. Specifically, the encoder processes $\mathbf{H}_A^f$ and $\mathbf{H}_B^g$, while the decoder processes $\mathbf{Q}_A$ and $\mathbf{H}_A^g$. The latents $\mathbf{H}_A^g$ and $\mathbf{H}_B^g$ serve as the initial tokens in the sequence, enabling both the encoder and decoder to operate over a sequence of length $T + 1$. The output sequence's final $T$ frames are represented as $\hat{\mathbf{Q}}_B$.

Due to the lack of paired ground-truth data, we employ the unsupervised method described by Lim, Chang, and Choi [15]. Our network utilizes four loss functions for training: reconstruction loss, DMI consistency loss, adversarial loss, and end-effector loss. Supervision signals are derived from the source motion. We maintain geometric interactions by aligning the source DMI field $\mathbf{D}_A$ with the target DMI field $\hat{\mathbf{D}}_B$. The target DMI field $\hat{\mathbf{D}}_B$ is generated by first applying sensor forward kinematics to $\hat{\mathbf{Q}}_B$, followed by selecting sensor pairs using the target sparse DMI mask $\mathbf{M}_{\text{tgt}} \in \mathbb{R}^{S \times S}$. This mask, $\mathbf{M}_{\text{tgt}}$, is derived by excluding invalid sensors of the target character from $\mathbf{M}_{\text{src}}$. The DMI consistency loss is quantified as the cosine similarity loss between pair-wise relative positions in $\hat{\mathbf{D}}_B$ and $\mathbf{D}_A$:

$$\mathcal{L}_{\text{dmi}} = -\frac{1}{T} \sum_{t=1}^{T} \sum_{k=1}^{K} \sum_{l=1}^{L} c(k,l) \frac{\mathbf{d}_A^{t,k,l} \cdot \hat{\mathbf{d}}_B^{t,k,l}}{||\mathbf{d}_A^{t,k,l}||_2 \cdot ||\hat{\mathbf{d}}_B^{t,k,l}||_2}, \tag{5}$$

where $c(k, l)$ takes the value 1 if sensor pair $(k, l)$ is valid in both $\mathbf{M}_{\text{src}}$ and $\mathbf{M}_{\text{tgt}}$, and 0 otherwise. The reconstruction loss serves as a regularization mechanism to minimize motion alterations during retargeting, defined as follows:

$$\mathcal{L}_{\text{rec}} = ||\hat{\mathbf{Q}}_{\text{B}} - \mathbf{Q}_{\text{A}}||_2^2. \tag{6}$$

To facilitate realistic motion retargeting, a discriminator, denoted as $\delta(\cdot)$, is employed. The adversarial loss is subsequently defined as:

$$\mathcal{L}_{\text{adv}} = \mathbb{E}_{\mathbf{Q} \sim p_{\text{real}}}[\log \delta(\mathbf{Q})] + \mathbb{E}_{\mathbf{Q} \sim p(\hat{\mathbf{Q}}_{\text{B}})}[\log(1 - \delta(\mathbf{Q}))]. \tag{7}$$

We observed that the global orientation of end-effectors significantly influences user experience. Consequently, we introduced an end-effector loss to promote consistent orientations of end-effectors in the retargeted motion.

$$\mathcal{L}_{\text{ef}} = \frac{1}{T|\mathcal{X}|} \sum_{t=1}^{T} \sum_{i \in \mathcal{X}} ||R(\mathbf{Q}_{\text{A}}^t, i) - R(\hat{\mathbf{Q}}_{\text{B}}^t, i)||, \tag{8}$$

where $R(\cdot)$ transforms local joint rotations into global rotations for joint $i$ along the kinematic chain and $\mathcal{X}$ represents the set of end-effectors. Our *MeshRet* is trained by:

$$\mathcal{L}_{\text{total}} = \lambda_{\text{rec}}\mathcal{L}_{\text{rec}} + \lambda_{\text{dmi}}\mathcal{L}_{\text{dmi}} + \lambda_{\text{adv}}\mathcal{L}_{\text{adv}} + \lambda_{\text{ef}}\mathcal{L}_{\text{ef}}. \tag{9}$$

## 4 Experiments

### 4.1 Settings

**Datasets** We trained and evaluated our method using the Mixamo dataset [2] and the newly curated *ScanRet* dataset. We downloaded 3,675 motion clips performed by 13 cartoon characters from the Mixamo dataset contains, while the *ScanRet* dataset consists of 8,298 clips executed by 100 human actors. Notably, the Mixamo dataset frequently features corrupted data due to interpenetration and contact mismatches. To overcome these issues, we created the *ScanRet* dataset, which provides detailed contact semantics and improved mesh interactions, with each clip being scrutinized by human animators. The training set comprises 90% of the motion clips from both datasets, involving nine characters from Mixamo and 90 from *ScanRet*. Our experiments tested the motion retargeting capabilities between cartoon characters and real humans, aligning closely with typical retargeting workflows. During inference, we adopted four data splits based on character and motion visibility: unseen character with unseen motion (UC+UM), unseen character with seen motion (UC+SM), seen character with unseen motion (SC+UM), and seen character with seen motion (SC+SM), as delineated by Zhang et al. [32]. We present the average results across these splits. Additional details available in Appendix A.

**Implementation details** The hyper-parameters $\lambda_{\text{rec}}$, $\lambda_{\text{dmi}}$, $\lambda_{\text{adv}}$, $\lambda_{\text{ef}}$, and $L$ were empirically set to $1.0, 5.0, 1.0, 1.0$, and $20$, respectively. We use $\{0, 1, \cdots, N_{\text{body}} - 1\} \times \{0, 0.25, 0.5, 0.75\} \times \{0, 0.5\pi, \pi, 1.5\pi\}$ as the SCS semantic coordinates set, where $N_{\text{body}} = 18$ is the number of body bones and $\times$ represents the Cartesian product. We employed the Adam optimizer [13] with a learning rate of $10^{-4}$ to optimize our network. The training process required 36 epochs. For further details, please refer to Appendix C.

**Evaluation metrics** We assess the effectiveness of our method through three metrics: joint accuracy, contact preservation, and geometric interpenetration. Joint accuracy is quantified by calculating the Mean Squared Error (MSE) between the retargeted joint positions and the ground-truth data provided by animators in ScanRet. This analysis considers both global and local joint positions, normalized by the character heights. Contact preservation is evaluated by measuring the Contact Error, defined as the mean squared distance between sensors that were originally in contact in the source motion clip. Geometric interpenetration is determined by the ratio of penetrated limb vertices to the total limb vertices per frame. Further details are available in Appendix B.

### 4.2 Comparison with state-of-the-arts

**Qualitative results** Figure 4 demonstrates the performance of skinned motion retargeting across characters with diverse body shapes, where the motion sequences are novel to the target characters

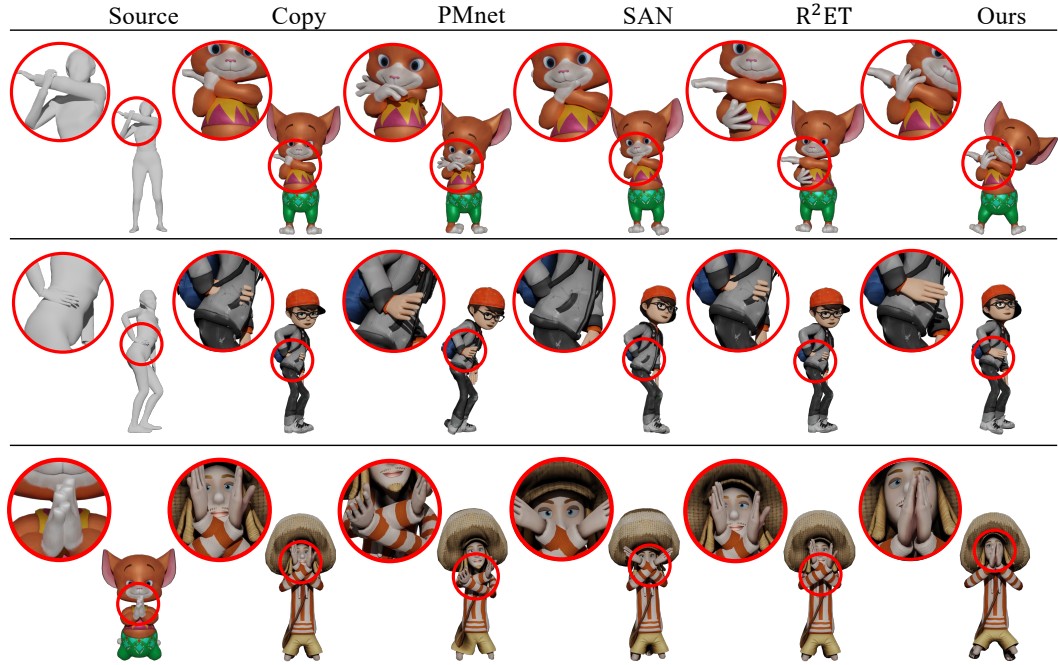

Figure 4: Qualitative comparison with baseline methods. Our method ensures precise contact preservation and minimal geometric interpenetration.

during training. Most baseline methods, except R$^2$ET [32], fail to consider the geometry of characters, leading to significant geometric interpenetration and contact mismatches. Unlike these methods, R$^2$ET [32] includes a geometry correction phase after skeleton-aware retargeting. However, this creates a conflict between the two stages, resulting in oscillations in R$^2$ET's outcomes, which manifest as alternating contact misses and severe interpenetrations, as shown in the first two rows. Additionally, these oscillations appear variably across different frames within the same motion clip, producing jittery motion, as illustrated in Figure 1 and Figure 8. A further limitation of R$^2$ET is its neglect of hand contacts. In contrast, our method employs the innovative DMI field to preserve such detailed interactions, such as those observed in the "Praying" pose in the third row.

Table 1: Quantitative comparison between our method and state-of-the-arts. Mixamo+ represents the mixed dataset of Mixamo and *ScanRet*. MSE$^{lc}$ denotes the local MSE.

| Metric | MSE↓ | MSE$^{lc}$↓ | Contact Error↓ | | Penetration(%)↓ | |
|---|---|---|---|---|---|---|
| **Dataset** | *ScanRet* | *ScanRet* | Mixamo+ | *ScanRet* | Mixamo+ | *ScanRet* |
| Source | - | - | - | 0.234 | 3.04 | 1.37 |
| Copy | 0.026 | 0.006 | 1.702 | 0.387 | 5.26 | 2.16 |
| PMnet [15] | 0.130 | 0.029 | 2.716 | 0.890 | 5.23 | 2.23 |
| SAN [1] | 0.049 | 0.011 | 2.432 | 0.627 | 4.95 | 1.72 |
| R$^2$ET [32] | 0.063 | 0.017 | 2.209 | 0.589 | 4.21 | 2.01 |
| Ours$_{cls}$ | 0.048 | 0.013 | 0.800 | 0.426 | **3.35** | 1.73 |
| Ours$_{far}$ | **0.045** | 0.010 | 1.642 | 0.610 | 5.37 | 1.77 |
| Ours$_{dm}$ | 0.048 | 0.010 | 2.568 | 0.797 | 4.69 | 1.78 |
| Ours | 0.047 | **0.009** | **0.772** | **0.284** | 3.45 | **1.59** |

**Quantitative results** Table 1 presents a comparison between our methods and state-of-the-arts. We initially measure the joint location error using MSE and MSE$^{lc}$ on *ScanRet*. The ground truth in *ScanNet* is established by human animators. Our observations indicate that human animators typically retarget motions by initially replicating joint rotations and subsequently modifying frames

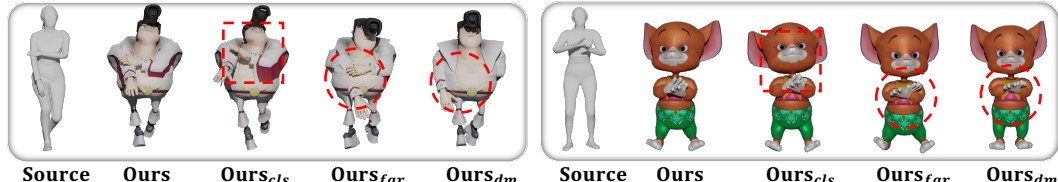

| Source | Ours | Ours$_{cls}$ | Ours$_{far}$ | Ours$_{dm}$ | | Source | Ours | Ours$_{cls}$ | Ours$_{far}$ | Ours$_{dm}$ |

Figure 5: Qualitative comparison of ablation studies. A red circle highlights areas of interpenetration, while a red rectangle identifies errors in non-contact semantics.

that display incorrect interactions. Conversely, our method modifies the entire motion sequence, resulting in a higher MSE compared to the Copy strategy. Nevertheless, MSE remains a valuable auxiliary reference. In comparison to PMnet [15], R$^2$ET [32], and SAN [1], our method achieves MSE reductions of 65%, 29%, and 8%, respectively. These results demonstrate that our approach more closely aligns with the outputs produced by human animators.

As shown in Table 1, PMnet [15] and SAN [1], exhibit high interpenetration ratios and contact errors due to their neglect of character geometries. R$^2$ET [32] effectively reduces interpenetration through a geometry correction stage; nonetheless, it still encounters high contact errors stemming from conflicts between the retargeting and correction stages. Our approach explicitly models geometry interactions and thereby achieves low contact error and penetration ratio, illustrating the effectiveness of our proposed *MeshRet* in generating high-quality retargeted motions with detailed contact semantics and smooth mesh interactions. Additionally, we observe that retargeting using the mixed Mixamo+ dataset is more challenging than with the *ScanRet* dataset, attributable to significant body shape variations between cartoon characters and real person characters.

## 4.3 Ablation Studies

We conducted ablation studies to demonstrate the significance of pairwise interaction feature selection and the implementation of DMI similarity loss. Initially, we evaluated the performance of a model trained exclusively with the nearest $L$ sensor pairs, denoted as Ours$_{cls}$, and another model trained solely with the farthest $L$ sensor pairs, referred to as Ours$_{far}$. As indicated in Table 1 and Figure 5, Ours$_{far}$ compromises contact semantics and leads to significant interpenetration, while Ours$_{cls}$ also exhibits inferior performance. This outcome suggests that proximal sensor pairs are essential for minimizing interpenetration and preserving contact, whereas distal pairs provide insights into the non-contact spatial relationships among body parts. Further, we investigated the effect of incorporating a distance matrix loss, as proposed by Zhang et al. [32], on our sensor pairs, designated as Ours$_{dm}$. The results imply that the distance matrix loss fails to yield meaningful supervisory signals, likely because distance is non-directional and insufficient to discern the relative spatial positions among numerous sensors.

Table 2: Human preferences between our method and baselines.

| Methods | Semantics Preservation | Contact Accuracy | Overall Quality |
|---------|------------------------|------------------|-----------------|
| Copy | 20.7% v.s. **79.3%(Ours)** | 22.7% v.s. **77.3%(Ours)** | 18.7% v.s. **81.3%(Ours)** |
| PMnet [15] | 2.7% v.s. **97.3%(Ours)** | 5.3% v.s. **94.7%(Ours)** | 1.3% v.s. **98.7%(Ours)** |
| SAN [1] | 9.3% v.s. **90.7%(Ours)** | 15.3% v.s. **84.7%(Ours)** | 7.3% v.s. **92.7%(Ours)** |
| R$^2$ET [32] | 14.6% v.s. **85.4%(Ours)** | 16.0% v.s. **84.0%(Ours)** | 13.3% v.s. **86.7%(Ours)** |

## 4.4 User study

We conducted a user study to assess the performance of our MeshRet model in comparison with the Copy strategy, PMnet [15], SAN [1], and R$^2$ET [32]. Fifteen sets of motion videos were presented to participants, each consisting of one source skinned motion and five anonymized skinned results. Participants were requested to rate their preferences based on three criteria: semantic preservation, contact accuracy, and overall quality. Users were recruited from Amazon Mechanical Turk [3],

resulting in a total of 600 comparative evaluations. As indicated in Table 2, approximately 81% of the comparisons favored our results. Details can be found in Appendix D

# 5  Conclusion

We introduce a novel framework for geometric interaction-aware motion retargeting, named *MeshRet*. This framework explicitly models the dense geometric interactions among various body parts by first establishing a dense mesh correspondence between characters using semantically consistent sensors. We then develop a unique spatio-temporal representation, termed the DMI field, which adeptly captures both contact and non-contact interactions between body geometries. By aligning this DMI field, *MeshRet* achieves detailed contact preservation and seamless geometric interaction. Performance evaluations using the Mixamo dataset and our newly compiled *ScanRet* dataset confirm that *MeshRet* offers state-of-the-art results.

**Limitations**   The primary limitation of *MeshRet* is its dependence on inputs with clean contact; motion clips exhibiting severe interpenetration yield poor outcomes. Consequently, it is unable to process noisy inputs effectively. Refer to Figure 12 and Figure 13 for failure cases under noisy inputs. Future efforts will focus on enhancing its robustness to noisy data. Additionally, SCS extraction can be compromised by noisy meshes, particularly those with complex clothing. A potential solution is to employ a Laplacian-smoothed proxy mesh for SCS extraction. Lastly, the method cannot handle characters with missing limbs.

## Acknowledgments and Disclosure of Funding

This work is supported by the National Key R&D Program of China under Grant No. 2024QY1400, the National Natural Science Foundation of China No. 62425604, and the Tsinghua University Initiative Scientific Research Program. Mike Shou does not receive any funding for this work.

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

# A  Dataset Details

***ScanRet* details**   The primary motivation for collecting the *ScanRet* dataset stemmed from two main concerns. First, the data quality in the Mixamo [2] dataset was relatively low, suffering from significant issues such as interpenetration and contact mismatch. Second, the Mixamo dataset exclusively contained cartoon characters, whose body type distributions differed markedly from those of real human motion capture actors. In response, we developed the *ScanRet* dataset. We recruited 100 participants, evenly split between males and females, representing common ranges of height and BMI. Each participant underwent a 3D scan to create a T-pose mesh. We intentionally did not collect texture information for the body or face to protect privacy. Subsequently, we used motion capture equipment to build a library of 83 actions characterized by extensive physical contact. We enlisted human animators to map each action onto the 100 T-pose meshes, ensuring both semantic integrity and correct physical contact were maintained. All participants and animators received fair compensation. After discarding some invalid data, we compiled a total of 8,298 motion data entries. The ScanRet dataset is designed to simulate data obtained from real human motion capture, such as the MoSh [20, 17] algorithm, thus enhancing the realism of our evaluation process in the context of actual animation production workflows.

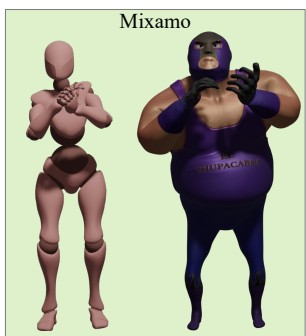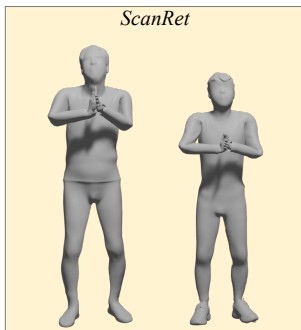

Figure 6: Left: Characters of varying body types in the Mixamo dataset do not always maintain reasonable hand contact during clapping actions. Right: In our ScanRet dataset, characters of diverse body types consistently maintain appropriate hand contact while performing the same clapping actions.

**Data splits**   We collected motion data for 13 characters from the Mixamo website, totaling 3,675 motion sequences, with each character having approximately the same number of sequences. The characters are: Aj, Amy, Kaya, Mousey, Ortiz, Remy, Sporty Granny, Swat, The Boss, Timmy, X Bot, and Y Bot. Among them, Ortiz, Kaya, X Bot, and Amy were not encountered by the network during training. Overall, our training set included motion data for 9 Mixamo characters and 90 randomly selected characters from the ScanNet dataset, where 90% of the motion sequences was randomly chosen from both datasets. Details regarding the train/test split for specific motion sequences and characters are provided in the code.

# B  Evaluation metric details

We evaluate the performance of our method from three perspectives: joint accuracy, contact preservation, and geometric interpenetration. In terms of joint accuracy, we calculate the Mean Squared Error (MSE) between the ground-truth joint positions $X_{gt}$ and the retargeted joint positions $\hat{X}$, normalized by the character's height $h$:

$$MSE = \frac{1}{h}||X_{gt} - \hat{X}||_2^2 \tag{10}$$

Previous work [32] assessing the accuracy of self-contact measurements merely utilized the distance between hand vertices and the body surface to determine contact presence. Such experimental metrics fail to accurately reflect the precision of the contact location. Therefore, we adopted a metric similar to the vertex contact mean squared error (MSE) proposed by Villegas et al. [28], termed "Contact Error". Specifically, we first identified sensor pairs where the distance between hand and body sensors

in the source action was less than the arm's diameter $d_{src}$. We then located the same sensor pairs in the retargeted motion. If the distance between these sensor pairs in the retargeted motion exceeded that in the source action, we calculated the MSE of the distance differences; otherwise, the contact error was zero. The formula is as follows:

$$\text{Contact Error} = \begin{cases} (||\frac{\mathbf{d}_A^{t,k,l}}{R_A}||_2 - ||\frac{\hat{\mathbf{d}}_B^{t,k,l}}{R_B}||_2)^2, & \text{if} ||\frac{\mathbf{d}_A^{t,k,l}}{R_A}||_2 > ||\frac{\hat{\mathbf{d}}_B^{t,k,l}}{R_B}||_2 \\ 0, & \text{otherwise}, \end{cases} \quad (11)$$

where $\mathbf{d}_A^{t,k,l}$ indicates the contact sensor pairs with $||\mathbf{d}_A^{t,k,l}||_2 < d_{src}$, while $R_A$ and $R_B$ represent the radius of each character's arms.

For geometric interpenetration, we assess the percentage of interpenetration, calculated as the ratio of penetrated vertices to the total vertices per frame. A lower ratio signifies reduced interpenetration. In our evaluation, we calculate the interpenetration ratio between arms (including hands) and the body.

$$\text{Penetration} = \frac{\text{Number of penetrated arm vertices}}{\text{Total number of arm vertices}}. \quad (12)$$

## C  Implementation Details

**SCS details**    As introduced in Section 3.2, we establish semantic correspondences between character meshes with different topologies using semantically consistent sensors. Specifically, given the semantic coordinates $(b, l, \phi)$ of a sensor, we can identify semantically consistent sensor positions on the meshes of different roles and obtain the feature vectors of the sensors. This process is detailed in Algorithm 1.

---

**Algorithm 1:** Derive Semantically Consistent Sensors from Semantic Coordinate

---

**Input:** Mesh $\mathbf{O}$, joint locations $\mathbf{J} \in \mathbb{R}^{N \times 3}$, bone index $b \in \{0, 1, \cdots, N\}$, origin parameter
      $l \in [0, 1)$, direction parameter $\phi \in [0, 2\pi)$
**Output:** Sensor feature $\mathbf{s} \in \mathbb{R}^{4 \times 3}$
$i_{\text{parent}} \leftarrow \text{bone\_parent\_joint}(b)$, $i_{\text{child}} \leftarrow \text{bone\_child\_joint}(b)$;
$\mathbf{x}_{\text{parent}} \leftarrow \mathbf{J}[i_{\text{parent}}]$, $\mathbf{x}_{\text{child}} \leftarrow \mathbf{J}[i_{\text{child}}]$;
$\mathbf{o} \leftarrow (1 - l)\mathbf{x}_{\text{parent}} + l\mathbf{x}_{\text{child}}$ ;                                    /* Ray origin */
$\mathbf{d}_{\text{forward}} \leftarrow \text{forward\_direction}(\mathbf{O})$ ;                      /* Face forward direction */
$\mathbf{d}_{\text{bone}} \leftarrow \text{normalize}(\mathbf{x}_{\text{child}} - \mathbf{x}_{\text{parent}})$ ;              /* Bone unit direction vector */
$\mathbf{d}_{\text{other}} \leftarrow \mathbf{d}_{\text{forward}} \times \mathbf{d}_{\text{bone}}$;
$\mathbf{n} \leftarrow \cos(\phi)\mathbf{d}_{\text{forward}} + \sin(\phi)\mathbf{d}_{\text{other}}$ ;                              /* Ray direction */
$\mathbf{B} \leftarrow \text{bone\_mesh}(\mathbf{O}, b)$ ;                                    /* Bone associated mesh */
$\mathbf{r} \leftarrow \text{ray}(\mathbf{o}, \mathbf{n})$;
$\mathbf{p} \leftarrow \text{ray\_mesh\_intersection}(\mathbf{B}, \mathbf{r})$;
**if** $\mathbf{p} \neq \emptyset$ **then**
   |    $\mathbf{t} \leftarrow \text{tangent\_matrix}(\mathbf{x}_{\text{p}}, \mathbf{B})$;
   |    $\mathbf{s} \leftarrow \text{concat}(\mathbf{p}, \mathbf{t})$;
**else**
   |    $\mathbf{s} \leftarrow \mathbf{0}$;
**end**

---

**Network architecture**    The network architectures of both our DMI Encoder and Geometry Encoder resemble the structure of PointNet. However, since all our data is inherently situated within the canonical space, we have eliminated the T-Net from PointNet to reduce network complexity. Before being input into the encoder, sensor features pass through a sensor group embedding layer, which converts the bone index $b$ into an 8-dimensional embedding vector. This embedding vector is updated during training. The Geometry Encoder consists of six PointNet layers with $D_{\text{model}}$ set at 256, and there is a distinct Geometry Encoder for the body, head, arms, and legs. The DMI Encoder comprises a per-sensor encoder and a per-frame encoder, each built with six PointNet layers, with each interaction pair having its own encoder. Specific interaction pairs include: [(Left Arm), (Right Arm, Head, Torso)], [(Right Arm), (Left Arm, Head, Torso)], [(Left Leg), (Right Leg, Torso)], and [(Right Leg), (Left Leg, Torso)]. The Motion Encoder is a multilayer perceptron (MLP). Both the

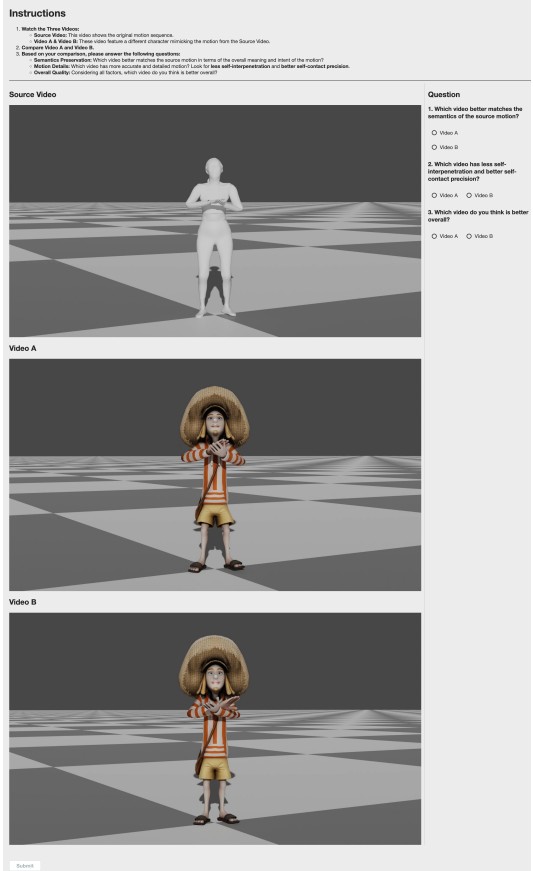

Figure 7: User interface presented to participants during the user study.

Transformer Encoder and Transformer Decoder have eight layers, with the number of heads set to four and the feed-forward size to 256. Between the Transformer Encoder and Transformer Decoder, we employ an alignment mask proposed by Fan et al. [6], which ensures that each frame feature in the decoder attends only to the corresponding DMI frame and initial token, thereby aligning the network's output motion sequence with the input features.

**Training details**    We implemented our network using PyTorch [23], running on a machine equipped with an NVIDIA RTX A6000 GPU and an AMD EPYC 9654 CPU. The dataset was uniformly processed at a frame rate of 30 fps. During training, we randomly clipped a sequence of 30 frames from the dataset. The target character was set to be the same as the source character with a 50% probability, and different with a 50% probability, selected randomly from the dataset. On our system, training for 36 epochs required approximately 40 hours. During inference, our *MeshRet* model can achieve performance exceeding 30 fps.

# D    User study details

We recruited participants via the Amazon Mechanical Turk [3] platform to partake in a user study. As shown in Figure 7, during each session, subjects were presented with one source video and two retargeted motion videos: Video A and Video B. Participants were asked to watch all three videos and then compare Video A and Video B. At the conclusion of the viewing, they were requested to answer the following three questions:

1. Which video better matches the source motion in terms of the overall meaning and intent of the motion?

2. Which video has more accurate and detailed motion? Look for less self-interpenetration and better self-contact precision.

3. Considering all factors, which video do you think is better overall?

For each question answered, participants received a compensation of $0.04. We collected 600 comparison results in the end.

# E    Additional results

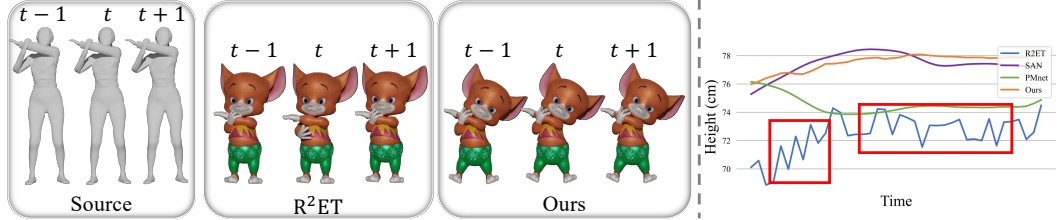

Figure 8: Left: We visualized three consecutive frames within an motion sequence. It is evident that while there was no jitter in the motion source, significant jitter occurred in the $t$-th frame of the R$^2$ET [32] results, which was not the case with our method. Right: We visualize the corresponding right-hand height for this segment of the sequence. The results indicate that the jitter in the R$^2$ET output was pronounced.

**Motion jitter comparison**    To better illustrate the jitter issue present in the results from the R$^2$ET [32] method, we visualized consecutive frames generated by R$^2$ET and our method in Figure 8, and provided a line graph depicting the variations in height of the right-hand joint over time. These results demonstrate that R$^2$ET is adversely affected by contradictions between skeletal retargeting and geometry correction phases, leading to significant motion jitter. In contrast, our method successfully avoids this problem.

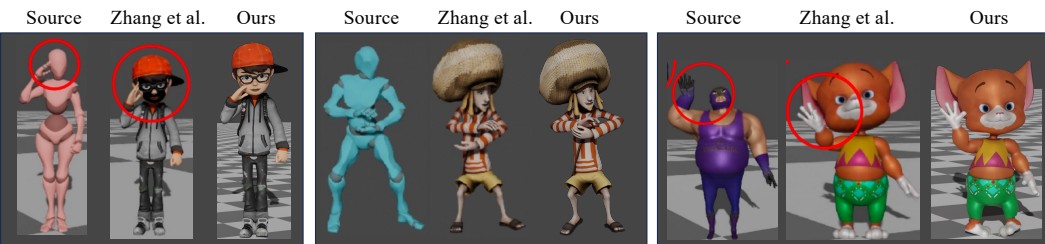

Figure 9: Qualitative comparison with Zhang et al. [30].

**Qualitative comparison with Zhang et al. [30]**    Since Zhang et al. [30] did not open-source their code, we were unable to conduct a complete and fair comparison of their method with ours in our experiments. However, we endeavored to locate several examples presented in their paper and applied our *MeshRet* to the same motion sequences. The comparative results are displayed in Figure 9. As observed in these examples, our method maintains the semantic integrity of the source motions, and it performs better in the Fireball case (the second motion sequence shown). This indicates that our method can achieve, and even surpass, the performance of their approach.

**Metrics across different data splits**    Tables 3 and 4 present the contact error and penetration ratio of our method compared to the baseline method across four different data splits. A consistent pattern observed is that performance improves for seen characters or motions. It is evident that our method outperforms the baseline across all data splits.

**Ablation studies on ratios of proximal sensor pairs**    The full approach can be considered a mixed version of Ours$_{far}$ and Ours$_{cls}$, utilizing an equal distribution of proximal and distal sensor pairs. To

Table 3: Contact errors of *MeshRet* and baselines across all data splits on Mixamo+.

| Metric | Contact Error↓ | | | |
|---|---|---|---|---|
| **Data Split** | UC+UM | SC+UM | UC+SM | SC+SM |
| Copy | 1.462 | 1.188 | 2.477 | 1.682 |
| PMnet [15] | 1.826 | 1.774 | 4.134 | 3.132 |
| SAN [1] | 1.416 | 1.181 | 4.229 | 2.902 |
| R$^2$ET [32] | 1.653 | 1.498 | 3.372 | 2.314 |
| Ours | **0.573** | **0.837** | **1.248** | **0.432** |

Table 4: Penetration ratios of *MeshRet* and baselines across all data splits on Mixamo+.

| Metric | Penetration(%)↓ | | | |
|---|---|---|---|---|
| **Data Split** | UC+UM | SC+UM | UC+SM | SC+SM |
| Copy | 1.57 | 4.16 | 5.78 | 9.56 |
| PMnet [15] | 1.43 | 4.20 | 5.71 | 9.56 |
| SAN [1] | 1.81 | 5.52 | 4.66 | 7.81 |
| R$^2$ET [32] | **1.54** | 4.66 | 4.92 | 5.71 |
| Ours | 1.55 | **2.63** | **4.60** | **5.04** |

better illustrate this balance, we provide additional experimental results by testing different ratios of proximal to distal sensor pairs. Table 5 compares our method's performance with varying percentages of proximal sensor pairs under the Mixamo+ setting. As the percentage of proximal sensor pairs decreases, the interpenetration ratio fluctuates mildly, while the contact error initially decreases and then increases. Finally, with no proximal pairs (equivalent to the "far" version), the performance drops significantly. In Figure 10, we present a qualitative comparison of our methods using different proximal sensor pair ratios. Except for the 100% Proximal version (equivalent to Ours$_{cls}$) and the 0% Proximal version (equivalent to Ours$_{far}$), our method demonstrates fair robustness to the proximal sensor ratio in the 25%-75% interval. Based on these results, we conclude that choosing 50% proximal sensor pairs strikes a reasonable balance for achieving good performance.

Table 5: Quantitative comparison between our methods with varing percentages of proximal sensor pairs under the Mixamo+ setting.

| Method | Contact Error↓ | Penetration(%)↓ |
|---|---|---|
| 100% Proximal Pairs | 0.800 | 3.35 |
| 75% Proximal Pairs | 0.909 | 3.61 |
| 50% Proximal Pairs | 0.772 | 3.45 |
| 25% Proximal Pairs | 0.781 | 3.29 |
| 0% Proximal Pairs | 1.642 | 5.37 |

**Ablation studies on different sensor arragements** We conducted further ablation studies on different sensor arrangements. Specifically, we evaluated the performance of a model trained with half the sample points in the $\phi$ space in SCS, denoted as Ours$_\phi$, and another model trained with half the sample points in the $l$ space in SCS, referred to as Ours$_l$. As shown in Table 6, Ours$_\phi$ compromises the interpenetration ratio, indicating that sufficient sample points in the space are crucial for avoiding interpenetration. We also found that both models introduce artifacts; please refer to Figure 11.

**Failure cases with noisy inputs** We provide resutls with clean and noisy inputs in Figure12 and Figure13. The results of *MeshRet* exhibit interpenetration with noisy inputs.

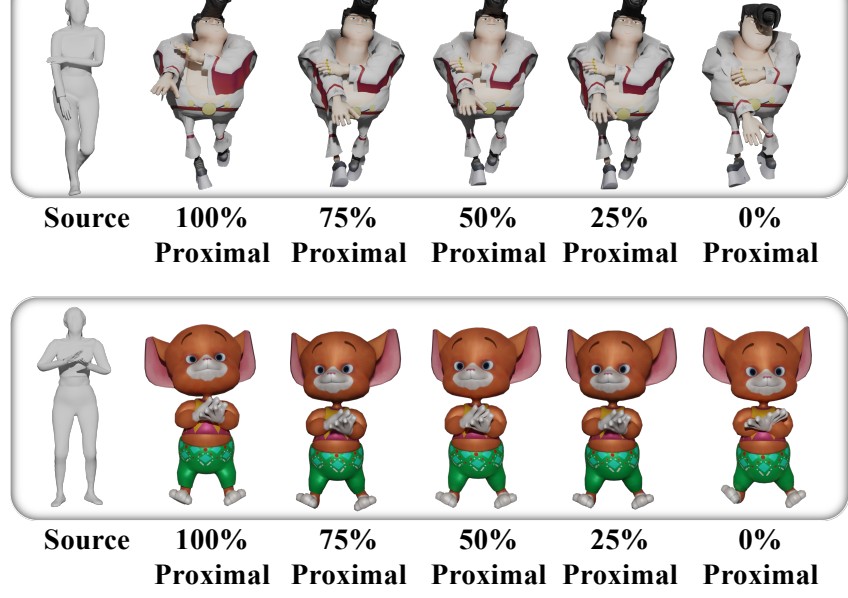

**Source**    **100% Proximal**    **75% Proximal**    **50% Proximal**    **25% Proximal**    **0% Proximal**

Figure 10: Qualitative results with different proximal sensor pair ratios.

Table 6: Quantitative comparison between methods with different sensor arrangements.

| Metric | MSE↓ | MSE$^{lc}$↓ | Contact Error↓ | | Penetration(%)↓ | |
|---|---|---|---|---|---|---|
| **Dataset** | *ScanRet* | *ScanRet* | Mixamo+ | *ScanRet* | Mixamo+ | *ScanRet* |
| Ours$_\phi$ | 0.052 | 0.011 | 0.793 | 0.410 | 3.90 | 1.60 |
| Ours$_l$ | 0.049 | 0.010 | 0.805 | 0.293 | 3.46 | **1.56** |
| Ours | **0.047** | **0.009** | **0.772** | **0.284** | **3.45** | 1.59 |

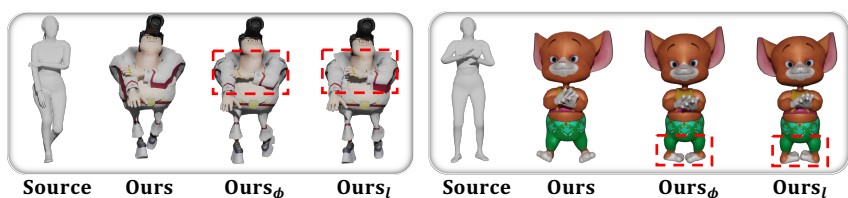

**Source**    **Ours**    **Ours$_\phi$**    **Ours$_l$**      **Source**    **Ours**    **Ours$_\phi$**    **Ours$_l$**

Figure 11: Qualitative comparison of additional ablation studies on sensor arrangements. The red rectangles identify artifacts introduced by different sensor arrangements.

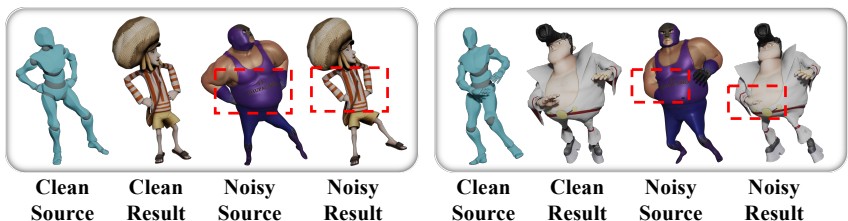

**Clean Source**   **Clean Result**   **Noisy Source**   **Noisy Result**    **Clean Source**   **Clean Result**   **Noisy Source**   **Noisy Result**

Figure 12: Qualitative results on the Mixamo dataset with clean and noisy inputs. A red rectangle indicates interpenetration.

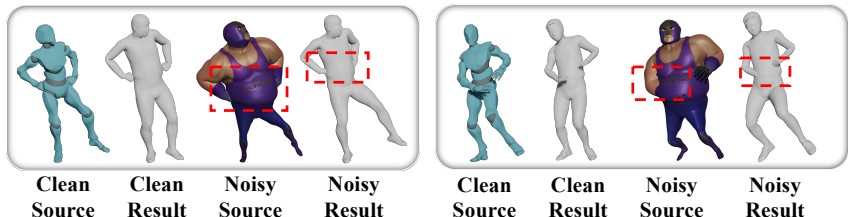

| Clean | Clean | Noisy | Noisy | | Clean | Clean | Noisy | Noisy |
| Source | Result | Source | Result | | Source | Result | Source | Result |

Figure 13: Qualitative results on the Mixamo dataset with *ScanRet* characters as targets. A red rectangle indicates interpenetration.

**More cases**    We present additional cases to validate the effectiveness of our *MeshRet*. Figures 14, 15, 16, and 17 depict four motion sequences retargeted from the source character to distinct target characters. These examples illustrate that our *MeshRet* is capable of generating high-quality motion sequences on target characters with diverse body shapes.

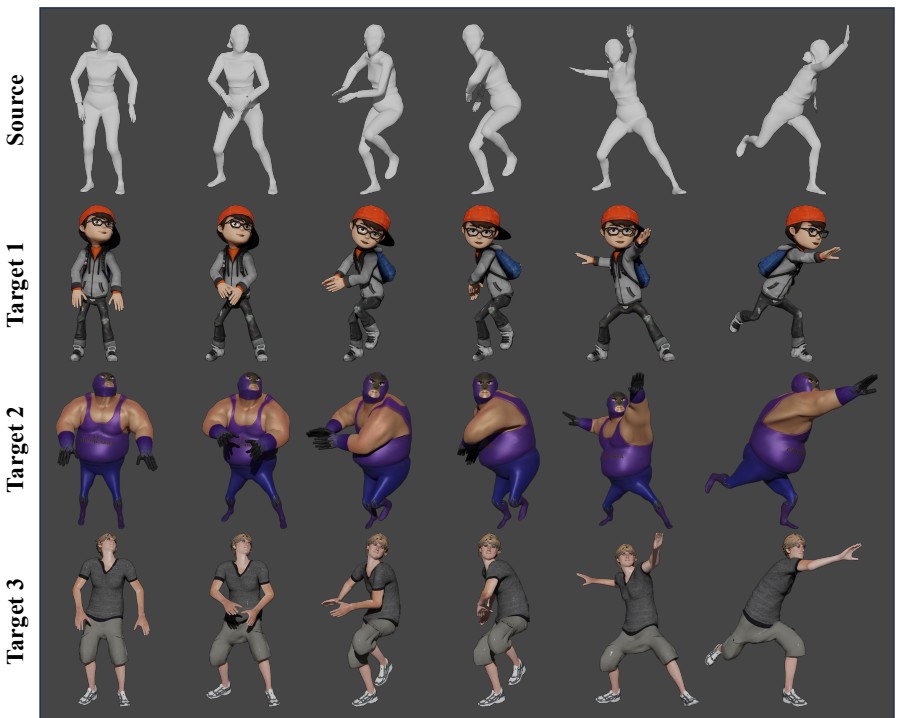

Figure 14: Snapshots of motion sequence 4 in *ScanRet*, retargeted from the source character to three distinct characters.

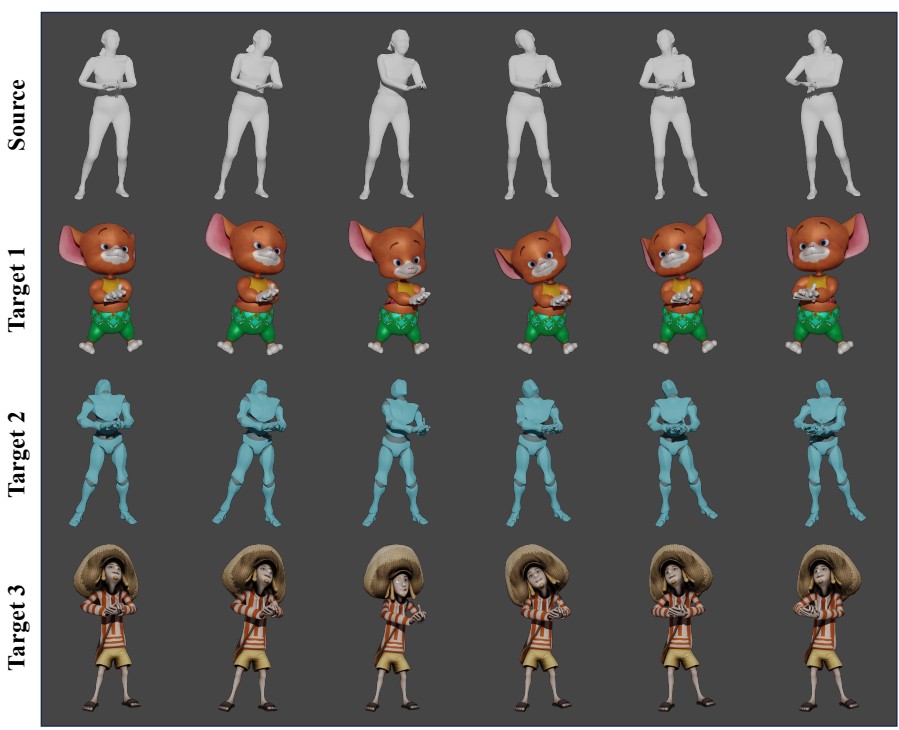

Figure 15: Snapshots of motion sequence 43 in *ScanRet*, retargeted from the source character to three distinct characters.

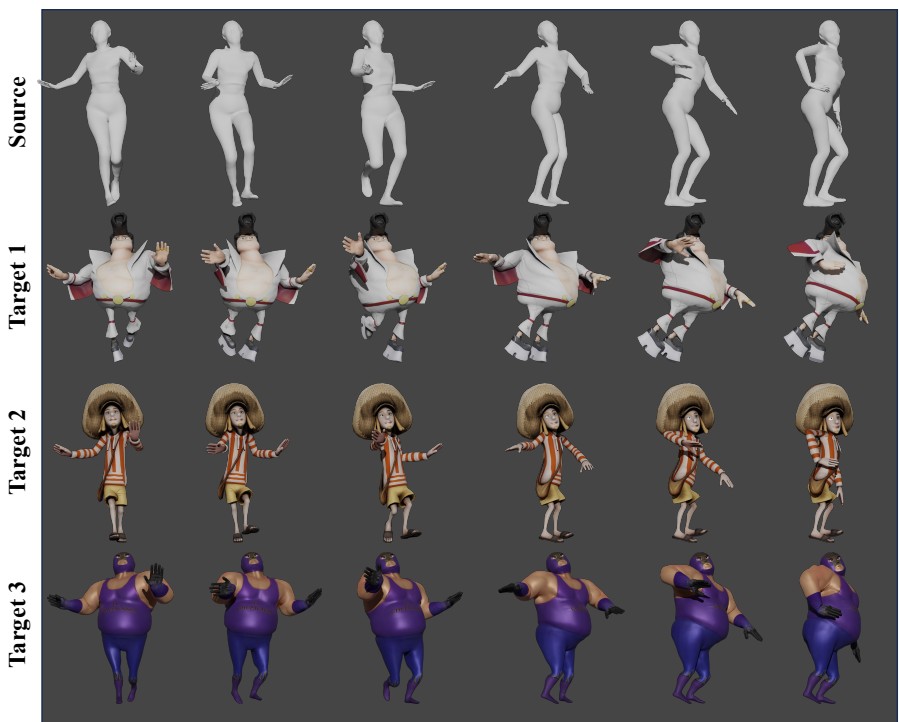

Figure 16: Snapshots of motion sequence 9 in *ScanRet*, retargeted from the source character to three distinct characters.

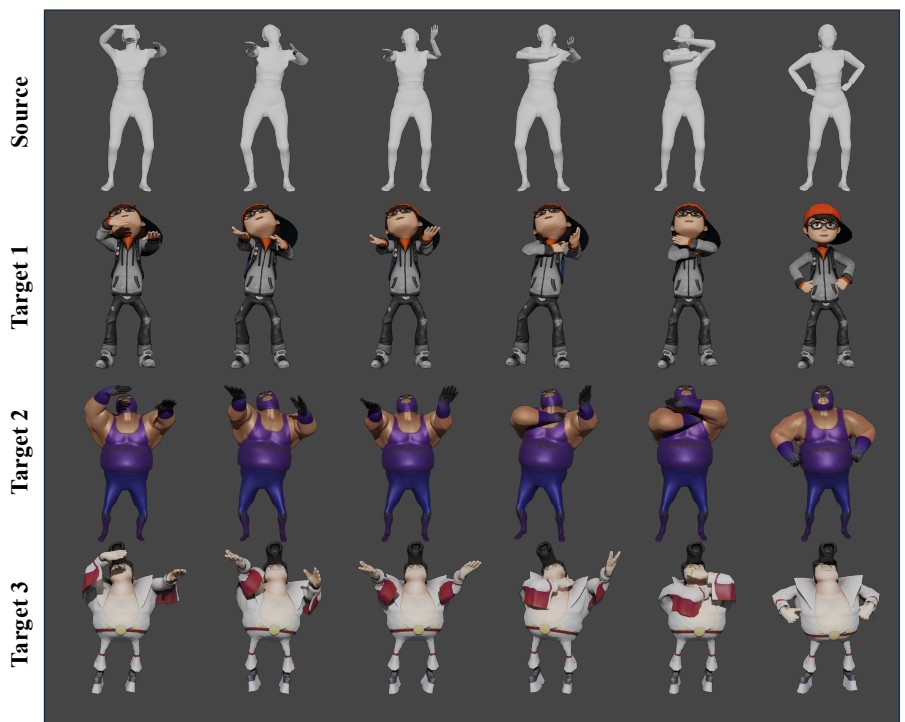

Figure 17: Snapshots of motion sequence 45 in *ScanRet*, retargeted from the source character to three distinct characters.

# F Broader impacts

Our work can provide animation professionals with enhanced results in motion retargeting, thereby alleviating their workload and increasing productivity in fields such as virtual reality, game development, and animation production. Regarding potential negative social impacts, we believe the likelihood of misuse of our work is minimal. This is because our work is situated in the midstream phase of the animation production pipeline, whereas privacy-invading forgeries, such as DeepFake, primarily occur during the downstream rendering phase.

