# OpenReview forum: "Skinned Motion Retargeting with Dense Geometric Interaction Perception"
_NeurIPS.cc/2024/Conference — NeurIPS 2024 spotlight_

### Official Review · Reviewer_YXjG · 2024-07-10

**Soundness:** 3
**Presentation:** 3
**Contribution:** 3
**Rating:** 6
**Confidence:** 4

**Summary:**

The paper proposes a data-driven method for character animation retargeting. The authors introduce Semantically Consistent Sensors (SCS), which are a set of surface points determined by casting rays from bones. Additionally, a Dense Mesh Interaction (DMI) field is proposed to measure the relative positions of sensors over time. They train a transformer that directly predicts motions of target characters, using source and target SCS, source motion, and source DMI. The training is conducted in an unsupervised manner, where the generated target DMI and target motion are matched to the source.

**Strengths:**

Many previous methods are compared, and the proposed method achieves state-of-the-art (SOTA) performance according to these comparisons.

**Weaknesses:**

The contact resolution relies on clean training data without penetrations. There is no mechanism to explicitly prevent penetrations.

**Questions:**

Why does Contact Error depend on absolute scale? What if the retargeted character has a significantly different scale from the source character?

What if the bones have different relative lengths? To achieve the same motion, they may require different orientations (Q). How diverse are the skeletons in your dataset? It would be interesting to see how the method performs when the relative bone lengths are different.

Does the method assume that all characters in the training dataset share the same bone system? What if I would like to use the method on a character system with a different number of bones and connectivities? Do I need to collect a new dataset? How many modifications to the neural networks are required?

**Limitations:**

The limitation that the retargeting quality depends on inputs with clean contact is discussed in the paper.

---

> ### Author Rebuttal · Authors · 2024-08-07
>
> We express our gratitude to Reviewer YXjG for acknowledging our extensive experiments and state-of-the-art performance. Your assistance in refining metric definitions is highly appreciated.
>
> # Q1: Contact Error depends on absolute scale.
>
> We agree that Contact Error should not depend on the absolute scale. Therefore, before calculating Contact Error, we normalize the DMI field by the limb radius of the character. Please refer to line 511 in `MeshRetCode/model/retnet.py` from our supplemental zip file for this detail.
>
> The precise definition of Contact Error should be:
> $$
> \text{Contact Error} = (||\frac{\mathbf{d}_A^{t,k,l}}{R_A}||_2 - ||\frac{\mathbf{d}_B^{t,k,l}}{R_B}||_2)^2 \quad \text{if}  ||\frac{\mathbf{d}_A^{t,k,l}}{R_A}||_2 > ||\frac{\mathbf{d}_B^{t,k,l}}{R_B}||_2 \quad \text{else} \quad 0,
> $$
> where $R_A$ and $R_B$ represent the radius of each character's arms.
>
> We apologize for not including this implementation detail in our paper and will correct this definition in our final manuscript.
>
> # Q2: How diverse are the skeletons in your dataset? Need for results on characters with different relative bone lengths.
>
> We plot the skeleton length distribution in Figure 15 of our rebuttal PDF. In the ScanRet dataset, the arm lengths of characters range from 51.9 cm to 67.0 cm, body heights range from 153.4 cm to 185.1 cm, and arm-height ratios range from 0.32 to 0.39. In the ScanRet dataset, the arm lengths of characters range from 35.8 cm to 134.2 cm, body heights range from 120.4 cm to 368.3 cm, and arm-height ratios range from 0.28 to 0.45.
>
> To see results on characters with different relative bone lengths, please refer to Figures 10-13 in our paper and the last two clips in our demo video from the supplemental zip file. In these results, our method performs well on characters with different relative bone lengths.
>
> # Q3: What if I would like to use the method on a character system with a different number of bones and connectivities?
>
> To process characters with different bone numbers, such as varying spine bone numbers, we can split or merge neighboring bones and then attach sensors to the modified virtual bone system. This approach requires no training or modification of the neural network. However, to process characters with different bone connectivities, we need to collect a new dataset and retrain the neural network. Additionally, some hyper-parameters, such as the dimensions of certain embedding layers, must be modified. Notably, previous state-of-the-art methods like R2ET and SAN also require characters to share a common skeleton topology.

---

> > ### Comment · Reviewer_YXjG · 2024-08-13
> >
> > Thank you for the response. I do not have further questions. Adding an experiment that can split or merge neighboring bones to handle different bone numbers would make the paper stronger. And please discuss the limitation as stated in Q3 in the revision.

---

> ### Author Response · Authors · 2024-08-13
> **Could you kindly review our response?**
>
> Dear Reviewer,
>
> Thank you once again for your feedback. As the rebuttal period is nearing its conclusion, could you kindly review our response to ensure it addresses your concerns?
>
> We appreciate your time and input.
>
> Sincerely,
>
> The Authors

---

> ### Author Response · Authors · 2024-08-14
> **Reply to Reviewer YXjG's further comment**
>
> Thank you for your valuable feedback and support. We are pleased to confirm that we have thoroughly addressed your concerns. Given the limited time remaining in the rebuttal phase and the necessity of refactoring some of our code, we will strive to provide experimental results on characters with varying bone numbers. If we are unable to complete this before the rebuttal deadline, we commit to including experiments with varying bone counts in our final manuscript, accompanied by a comprehensive discussion of the results. All the discussion about our method's limitation will also be included.

---

### Official Review · Reviewer_utNP · 2024-07-11

**Soundness:** 3
**Presentation:** 3
**Contribution:** 3
**Rating:** 6
**Confidence:** 4

**Summary:**

* The paper proposes a new motion retargeting method based on dense geometric information. This dense geometric information is gathered as dense interaction fields by pre-generated pseudo sensors, which are semantically embedded on the skin. The interaction fields are capable of representing the semantic information of different motions. Based on these ideas, the authors designed a new network to extract features from source motion, body shapes, and the newly designed interaction fields. The interaction fields are also used in the loss functions, together with GAN loss, reconstruction loss, and other loss functions. Experimental results show that the new method can effectively avoid contact loss and interpenetration while maintaining the semantics of the source motion.

* The main contribution is the MeshRet method, which includes the Semantically Consistent Sensors, Dense Mesh Interaction Fields, a new network architecture, and losses designed based on the previous two modules. The authors also collected a dataset named ScanRet to improve the quality of motion retargeting datasets.

**Strengths:**

* The method is new, inspired by previous works on pseudo sensors and motion retargeting. There was a skeletal distance matrix in R2ET, and the DMI proposed in this paper can be seen as an improved, dense version of the distance matrix. R2ET used the distance matrix solely for motion semantics, while MeshRet uses the DMI for both semantics and geometrics.

* DMI is a better representation of motion semantics compared to the distance matrix. The method has the potential to become the new SOTA.

* The overall organization of the paper is good.

**Weaknesses:**

Both qualitative and quantitative results illustrate the effectiveness of this method. However, there are some weaknesses in the experiments:

* The ablation study focuses on close/far sensor pairs and the distance matrix, but does not address other aspects, such as different sensor arrangements.

* The newly collected dataset, ScanRet, is only used as a source of motion. This raises the question of whether it can be used in conjunction with Mixamo.

* The paper compares its results with SAN, PM-Net, and R2ET, but does not include comparisons with other works, such as the 2021 paper "Contact-aware Retargeting of Skinned Motion."

* Both ScanRet and Mixamo datasets are used in training. It would be interesting to see the results if only the Mixamo dataset is used.

**Questions:**

Refer to weaknesses part.

**Limitations:**

* The authors mention that one limitation is poor outcomes when the input is noisy. However, they didn't provide any failed cases or examples under noisy conditions to clarify this limitation.

* Other limitations may include how to attach sensors for people with disabilities or different body shapes, and how to handle sensors for designed animated characters with more or fewer bones.

---

> ### Author Rebuttal · Authors · 2024-08-07
>
> We extend our appreciation to Reviewer utNP for recognizing the novelty of our method and the effectiveness of the proposed DMI field. We sincerely value your recommendation to include additional experimental results.
>
> # Q1: More ablation studies on different sensor arrangements.
>
> We provide additional ablation experiments here. We conducted further studies on different sensor arrangements. Specifically, we evaluated the performance of a model trained with half the sample points in the $\phi$ space in SCS, denoted as ${Ours}\_\phi$, and another model trained with half the sample points in the $l$ space in SCS, referred to as ${Ours}\_l$. As shown in the following table, ${Ours}_\phi$ compromises the interpenetration ratio, indicating that sufficient sample points in the $\phi$ space are crucial for avoiding interpenetration. We also found that both models introduce artifacts; please refer to Figure 14 in our rebuttal PDF.
>
> | **Metric**  | **MSE** &darr; | **MSE $^{lc}$** &darr; | **Contact Error** &darr; | **Contact Error** &darr; | **Penetration** (\%) &darr; | **Penetration** (\%) &darr; |
> |---|---|---|---|-----| -----| -----|
> | **Dataset** | ScanRet | ScanRet | Mixamo+ | ScanRet | Mixamo+ | ScanRet |
> | Ours $_{\phi}$ | 0.052 | 0.011 | 0.793 | 0.410 | 3.90 | 1.60 |
> | Ours $_{l}$ | 0.049 | 0.010 | 0.805 | 0.293 | 3.46 | **1.56** |
> | Ours | **0.047** | **0.009** | **0.772** | **0.284** | **3.45** | 1.59 |
>
>
> We will add the results to the final manuscript.
>
>
> # Q2: More results with ScanRet characters as target characters.
>
> We provide additional qualitative results using ScanRet characters as target characters. Please refer to Figure 17 in our rebuttal PDF. For quantitative results, the metrics in Table 1 are computed using both ScanRet characters and Mixamo characters as targets.
>
> # Q3: Comparison with "Contact-aware Retargeting of Skinned Motion."
>
> We value your advice on including more baseline methods. We intended to compare our method to "Contact-aware Retargeting of Skinned Motion," but we were unable to do so because the authors did not release their code, and some implementation details, such as the vertex correspondence feature, were not described in detail. Notably, the authors of R2ET did not compare their method to this one in their paper. In general, we have made our best effort to provide a comprehensive comparison with existing methods, including the currently open-sourced state-of-the-art method, R2ET.
>
> # Q4: The results only trained on the Mixamo dataset.
>
> We value your advice on including more results. The following table presents a comparison between our methods and state-of-the-art methods trained only on the Mixamo dataset.
>
> | **Method**  | **Contact Error** &darr; | **Penetration** (\%)&darr; |
> |---|---|-----|
> Copy | 2.77 | 4.07 |
> PMnet | 2.76 | 4.17 |
> SAN | 2.75 | 4.19 |
> R2ET | 2.81 | 4.15 |
> Ours | **2.07** | **3.47** |
>
> Because the Mixamo dataset does not always provide clean contact input, the performance of our method is affected. However, our method still achieves much lower Contact Error and Penetration Ratio compared to baseline methods. Please refer to Figure 16 in our rebuttal PDF for qualitative results.
>
>
> # L1: Failure cases under noisy conditions.
>
> We provide some failure cases in Figures 16 and 17 in our rebuttal PDF. We will include these in our final manuscript.
>
> # L2: How to attach sensors for disabled people with missing limbs and characters with different bone numbers?
>
> To process characters with different bone counts, such as varying spine bone numbers, we can first split or merge neighboring bones and then attach sensors to the modified virtual bone system. However, our method cannot process characters with missing limbs. We will address this limitation in our final manuscript and leave it to future work.

---

> > ### Comment · Reviewer_utNP · 2024-08-13
> >
> > Thanks for the detailed response by the authors to my previous comments and questions. I appreciate the effort you’ve put into addressing the issues I raised, and I’m pleased to see the improvements made in your revised manuscript. I have no further concerns or questions at this time.

---

> > > ### Author Response · Authors · 2024-08-13
> > > **Thanks to Reviwer utNP**
> > >
> > > Thank you for taking the time to review our paper and rebuttal. We will thoroughly address all the concerns raised by the reviewers by incorporating additional results and improving explanations for clarity in the final manuscript.

---

### Official Review · Reviewer_WToy · 2024-07-13

**Soundness:** 3
**Presentation:** 3
**Contribution:** 3
**Rating:** 7
**Confidence:** 4

**Summary:**

The author proposes MeshRet, a new solution that achieves geometry-aware motion retargeting across various mesh topologies in a single stage.

The author introduces semantically consistent sensors (SCS) and dense mesh interaction (DMI) field to guide the training of MeshRet, effectively achieving semantic alignment. SCS parametrically represents the relationship between points on the mesh surface and the skeleton. The DMI field expresses multi-frame motion features in a form similar to point cloud features.

The author presents a dataset called ScanRet, which is specifically tailored for evaluating motion retargeting techniques.

**Strengths:**

The author introduces the concept of SCS as a feature representation and proposes the use of DMI field for learning motion retargeting alignment. These technical points demonstrate innovation.

The article is well-articulated and easy to follow, particularly in the task definition section where the input-output relationship of the overall task and the relationships between different modules are clearly represented.

The experiments conducted in the article provide evidence of the effectiveness of the proposed method.

**Weaknesses:**

Overall, there are no significant drawbacks. However, the method of selecting $K \times L$ sensors from $S^2$ potential interacting sensors to balance complexity appears somewhat ad hoc.

Additionally, this method is only applicable to CG models and lacks robustness to real-world noise, a point the authors have also mentioned in the limitations section.

**Questions:**

Regarding the Semantically Consistent Sensors (SCS), how is the number S determined? Section 3.2 shows S, but later it is written as K. This part is rather confusing. Is it the total number versus the number after masking?

**Limitations:**

The author has already addressed the limitations of the method.

---

> ### Author Rebuttal · Authors · 2024-08-07
>
> We are grateful to Reviewer WToy for acknowledging our novel technical contributions and the effectiveness of our experimental evidence. We also appreciate your guidance in clarifying the notation in the DMI field description.
>
> # Q1: What are the number $S$ and $K$?
>
> We apologize for any confusion. The number $S$ is determined by the size of the SCS semantic coordinates set. Please refer to Line 257 for the specific semantic coordinates set used in our experiment, where $S = 288$. Your understanding of $K$ is correct. After masking the full DMI field $\overline{\textbf{D}} \in \mathbb{R}^{S \times S \times P}$ with $\mathbf{M}_{\text{src}} \in \mathbb{R}^{S \times S}$, we obtain the final DMI field $\mathbf{D} \in \mathbb{R}^{K \times L \times P}$, where $K$ represents the number of observation sensors in $\mathbf{D}$. We will include a more detailed explanation in the final version of our paper.

---

> > ### Comment · Reviewer_WToy · 2024-08-13
> >
> > Thanks to the authors for their detailed feedback.
> >
> > I don't have any new questions at this time. I hope the authors will consider all the concerns raised by the reviewers and address unclear expressions and other issues in the revised version.
> >
> > I will continue to maintain my current score.

---

> > > ### Author Response · Authors · 2024-08-13
> > > **Thanks to Reviewer WToy**
> > >
> > > Thank you for taking your time to go through our paper and rebuttal. We will definitely address all the concerns raised by reviewers by including additional results and enhancing explanations for clarity in our final manuscript.

---

### Official Review · Reviewer_yJ9L · 2024-07-13

**Soundness:** 3
**Presentation:** 3
**Contribution:** 3
**Rating:** 5
**Confidence:** 4

**Summary:**

To address the issues such as jittery, interpenetration, and contact mismatches issues caused in the motion retargeting task, this paper proposes a new framework named MeshRet, directly modeling the dense geometric interactions in motion retargeting. It initially establishes dense mesh correspondences using semantically consistent sensors (SCS), and develops a spatio-temporal representation named dense mesh interaction field (DMI), which can capture the contact and non-contact interactions between body parts. The proposed model, MeshRet, achieves SOTA performance on a public dataset,Mixamo, and a newly proposed dataset, ScanRet.

**Strengths:**

The proposed model, MeshRet, improves the geometric interaction-aware motion retargeting through a single pass. The proposed submodules, such as, SCS, DMI, can improve the contact and non-contact interaction semantics.

The SCS seems to be effective across various mesh topologies on different characters.

 The proposed model achieves SOTA performance on both a public dataset and a newly proposed dataset. The paper also provides good ablation studies, e.g., using nearest or farthest L sensors, and the idea of incorporating distance matrix loss.

The paper proposes a new dataset specifically tailored for assessing motion retargeting, it has contact semantics and ensures smooth mesh interaction. This dataset will benefit the research society. The paper also conducts a user study to demonstrate the superiority of the proposed model.

**Weaknesses:**

The ablation study suggests that the proximal sensor pairs are essential for minimizing interpenetration and preserving contact, whereas distal pairs provide insights into the non-contact spatial relationships among body parts. It’s a balancing problem to both maintain contact and at the same time to avoid penetration. Given a new unseen character for unknow motion retargeting, how to choose the “cls”, “far”, or “dm” version?

**Questions:**

Given a new unseen character for unknow motion retargeting, how to choose the “cls”, “far”, or “dm” version? This seems to be challenging as it lacks of prior knowledge of how much to enforce contact and how much to avoid interpenetration.

**Limitations:**

The model relies heavily on the accurate estimation of SCS, when the SCS extraction is affected by noisy input, complex clothes, it likes to decrease the performance.

---

> ### Author Rebuttal · Authors · 2024-08-07
>
> We extend our gratitude to Reviewer yJ9L for recognizing the novelty and state-of-the-art performance of our method. Your guidance on balancing proximal and distal sensor pairs has been particularly insightful.
>
> # Q1: How to choose the “cls”, “far”, or “dm” version?
>
> We appreciate your suggestion to discuss the balance problem between proximal and distal sensor pairs. In the ablation study presented in Section 4.3, the "dm" version ($Ours_{dm}$), the "cls" version ($\mathrm{Ours}_{cls}$), and the "far" version ($\mathrm{Ours}\_{far}$) all exhibit worse performance compared to the full approach (Ours). The full approach can be considered a mixed version of $\mathrm{Ours}\_{cls}$ and $\mathrm{Ours}\_{far}$, utilizing an equal distribution of proximal and distal sensor pairs. To better illustrate this balance, we provide additional experimental results by testing different ratios of proximal to distal sensor pairs. The following table compares our method's performance with varying percentages of proximal sensor pairs under the Mixamo+ setting. As the percentage of proximal sensor pairs decreases, the interpenetration ratio fluctuates mildly, while the contact error initially decreases and then increases. Finally, with no proximal pairs (equivalent to the "far" version), the performance drops significantly.
>
> | **Method**  | **Contact Error** &darr; | **Penetration** (\%) &darr; |
> |---|---|-----|
> 100% Proximal Pairs | 0.800 | 3.35 |
> 75% Proximal Pairs | 0.909 | 3.61 |
> 50% Proximal Pairs | 0.772 | 3.45 |
> 25% Proximal Pairs | 0.781 | 3.29 |
> 0% Proximal Pairs | 1.642 | 5.37 |
>
> In Figure 18 of our rebuttal PDF, we present a qualitative comparison of our methods using different proximal sensor pair ratios. Except for the 100% Proximal version (equivalent to $\mathrm{Ours}\_{cls}$) and the 0% Proximal version (equivalent to $\mathrm{Ours}\_{far}$), our method demonstrates fair robustness to the proximal sensor ratio in the 25%-75% interval.
>
> Based on these results, we conclude that choosing 50% proximal sensor pairs strikes a reasonable balance for achieving good performance. We will include the additional results in the final manuscript.
>
> As for the balance between enforcing contact and avoiding interpenetration, our method directly captures the interaction between mesh surfaces, as illustrated in Figure 3. Therefore, there is no explicit contradiction between preserving contact and avoiding interpenetration.
>
> # L1: The SCS extraction may be affected by noisy mesh input.
>
> Thank you for your valuable comments. Noisy mesh surfaces can indeed pose challenges. One potential solution is to adopt a Laplacian-smoothed proxy mesh of the character for the SCS extraction. We have not implemented this due to the limited time available during the rebuttal phase but plan to discuss this further in our final manuscript.

---

> ### Author Response · Authors · 2024-08-13
> **Could you kindly review our response?**
>
> Dear Reviewer,
>
> Thank you once again for your feedback. As the rebuttal period is nearing its conclusion, could you kindly review our response to ensure it addresses your concerns?
>
> We appreciate your time and input.
>
> Sincerely,
>
> The Authors

---

### Author Rebuttal · Authors · 2024-08-07

We sincerely thank all the reviewers for their valuable comments.

We extend our gratitude to Reviewer yJ9L for recognizing the novelty and state-of-the-art performance of our method. Your guidance on balancing proximal and distal sensor pairs has been particularly insightful.

We are grateful to Reviewer WToy for acknowledging our novel technical contributions and the effectiveness of our experimental evidence. We also appreciate your guidance in clarifying the notation in the DMI field description.

We extend our appreciation to Reviewer utNP for recognizing the novelty of our method and the effectiveness of the proposed DMI field. We sincerely value your recommendation to include additional experimental results.

We express our gratitude to Reviewer YXjG for acknowledging our extensive experiments and state-of-the-art performance. Your assistance in refining metric definitions is highly appreciated.

We have included additional figures in the global rebuttal PDF. Please refer to the PDF for further results.

---

### Decision · Program_Chairs · 2024-09-25

**Decision:**

Accept (spotlight)

**Comment:**

Claimed contributions
 - Character motion retargeting respecting geometric interactions by
    1. Build Semantically Consistent Sensors (SCS) by casting rays perpendicular to bones and finding intersections with the mesh
    2. Dense Mesh Interaction (DMI) Field storing relative positions and semantics (bone information) of sparsified pairwise SCS
    3. MesRet network with a transformer architecture taking the interaction features from the source DMI and geometry features from the source SCS to retarget poses to a target character respecting the source interactions

Strengths
 - Novel ideas, especially with the DMI and its sparsification strategy
 - Convincing results for the self-contact-aware retargeting

Weaknesses
 - Not much - the rebuttal and the discussions seem to resolve all concerns
 - I may ask the authors to optionally add a reference to "Spatial Relationship Preserving Character Motion Adaptation" by Ho et al, 2010.

The paper proposes a solid solution to an important problem in mapping skeletal animations from one character to another while respecting the self-contact. The authors were sincere in resolving all concerns raised by the reviewers. Incorporating the discussions and clarifications in the discussion phase in the final revision will give extra value to the paper. Thus, I propose accepting the paper as a spotlight.